# The burning island: Spatiotemporal patterns of fire occurrence in Madagascar

**Tristan Frappier-Brinton[1], Shawn M. Lehman[2]***

**1** Department of Biology, Duke University, Durham, North Carolina, United States of America, **2** Department of Anthropology, University of Toronto, Toronto, Ontario, Canada

* shawn.lehman@utoronto.ca

## Abstract

Anthropogenic fire use is widespread across Madagascar and threatens the island's unprecedented endemic biodiversity. The vast majority (96%) of lemur species are already threatened with extinction, and Madagascar has already lost more than 44% of its forests. Previous conservation assessments have noted the role of fire in the rampant deforestation and habitat degradation across Madagascar, but published, quantified data on fire use across the island are incredibly limited. Here, we present the first quantification of spatiotemporal patterns in fire occurrence across Madagascar using VIIRS satellite fire detection data. We assess which regions of Madagascar have the most prevalent fire use, how fire use is changing over time, and what this means for Madagascar's remaining forest ecosystems. An average of 356,189 fires were detected every year in Madagascar from 2012–2019, averaging 0.604 fires/km$^2$. Fire use was near-ubiquitous across the island, but was most prevalent in the western dry deciduous forests and succulent woodlands ecoregions. Fire frequency in the eastern lowlands was highest around the remaining humid rainforest, and fire frequency was increasing over time around much of the remaining humid and dry forest. We found that 18.6% of all remaining forest was within 500 m of a fire within a single year, and 39.3% was within 1 km. More than half of remaining forest was within 1 km of a fire in a single year in the dry deciduous forests, succulent woodlands, and mangroves ecoregions. However, fire frequency within national park protected areas was, on average, 65% lower than their surroundings. Only 7.1% of national park forest was within 500 m of a fire within one year, and 17.1% was within 1 km, suggesting that national parks are effective at reducing fire frequency in Madagascar's tropical forests.

## 1. Introduction

### 1.1 The issue of fire in the tropics

Fires are contributing to the destruction and degradation of tropical forests worldwide [1–3] and are only expected to become more severe with climate change [4, 5]. Even lush rainforests that used to experience natural fires once in hundreds to thousands of years are now burning regularly [1, 6–9]. Recent tropical forest fires in the Amazon Basin [9] and Southeast Asia [10]

**Funding:** SML, No grant number, Discovery Grant from the Natural Sciences and Engineering

Research Council of Canada, https://www.nserc-crsng.gc.ca/index_eng.asp. The funders had no role in study design, data collection and analysis, decision to publish, or preparation of the manuscript.

**Competing interests:** The authors have declared that no competing interests exist.

have illustrated fire's devastating effects on tropical forests ecosystems. Most native tree species in tropical forest are not adapted to withstand fires and suffer high immediate and delayed mortality [11, 12]. However, there are few data on the effects of fires on forest cover in Africa.

Africa is the most fire-threatened continent on Earth. Fires in Africa comprise 70% of the total annual burned area worldwide [13] and are responsible for 50% of global carbon emissions due to fire [8]. The impacts of forest fires are of particular concern in mega-diverse regions like Madagascar, which has some of the most concentrated endemic biodiversity across taxa of any nation [14, 15]: 80–90% of species in Madagascar are found nowhere else on Earth [16]. Most (96%) of Madagascar's 107 endemic lemur species are already threatened with extinction due to anthropogenic pressures and almost a third (31%) are considered critically endangered, the final classification before "extinct in the wild" [17]. It is estimated that 88% of species in Madagascar, including lemurs, cannot survive outside of forests [18], which makes the destructive impacts of fire on tropical forest a concern for both floral and faunal conservation in Madagascar.

Madagascar has lost more than 44% of its forests since 1953 [19]. While there is still debate on the extent of original forest before the 1950s, Madagascar may have already lost more than two thirds of its original forest cover [20]. Previous research has noted the widespread use of anthropogenic fires across Madagascar, primarily for agriculture land management [21, 22], and the destructive impacts fires have on Madagascar's already threatened forests [23–25]. However, there are few published studies that quantify patterns of fire occurrence across the island. In this paper, we present the first analysis to quantify the distribution and seasonal timing of fires across the entire island of Madagascar based on satellite VIIRS data.

## 1.2 Madagascar's fire regime

Fires in Madagascar are overwhelmingly started by humans [22, 26]. Fire exposure is now a major source of mortality for many Malagasy forest species [26], including species that evolved alongside natural fire occurrence [27]. The frequency of fires is now considerably higher than the frequency that these ecosystems evolved to cope with naturally, and humans have spread fire into habitats that were previously too wet or too dry to support burning [28]. Consequently, the fire regime across most of Madagascar is considered to be "Very Degraded", the worst possible classification [29].

The Malagasy people have typically used fire to clear forests for agricultural crops and to rejuvenate pastures for cattle [22]. Grassland fires are most common in the central plateau and western lowlands in the dry season (~May-November) [21]. Most grass fires are lit in the second half of the dry season to replace the standing lignified grasses with new growth shoots that are more nutritious for cattle [22]. However, many grassland fires spread into nearby forests, either intentionally or unintentionally, where they contribute to habitat degradation and the conversion of primary forest into secondary forest and, eventually, grassland [26]. Most fires occur in the late dry season, when they have higher intensity, making their spread into forests more likely and more destructive [26].

Intentional burning of forest to clear land for agriculture is particularly common in the eastern lowland rainforests, where more than 50% of original forests are believed to have been lost to slash-and-burn agriculture [19]. Forests are cut in the mid-late dry season, then residual biomass is burned near the end of the dry season (October-November) [22]. Slash-and-burn agriculture also threatens western dry forests for cropland conversion, particularly for the growing of maize [30, 31]. Most soils in Madagascar can only support crop growth for a few years before they must be abandoned and left to fallow so the soil nutrients can recover [32]. This fallowing cycle, combined with Madagascar's rapid population growth of 2.7% per year

[33], drives the continuing expansion of slash-and-burn agriculture into new lands, creating constant pressure on existing forests [34]. The time period between fallow cycles has also decreased from 8–15 years to a largely unsustainable 3–5 years, which is increasing fire frequency and accelerating the conversion of natural forests into secondary grassland [32]. Across the island, anthropogenic fires are also used for the herding of zebu cattle [35], charcoal production, particularly in the southwest of the island [31], and lime clay production, particularly in coastal mangrove regions [36].

### 1.3 Madagascar's ecoregions

Madagascar's topography has given rise to many unique ecoregions with different climates, species compositions, and susceptibilities to fire (Fig 1A). Here, we consider seven ecoregions based on the World Wildlife Fund ecoregion assessment [37]: Dry deciduous forests, Subhumid forests, Lowland forests, Succulent woodlands, Spiny thickets, Ericoid thickets, and Mangroves. The western dry deciduous forests are composed of forest fragments surrounded by grassland savannas [30]. These forests experience a long, pronounced dry season, which typically runs from May through November. The subhumid forest ecoregion is found along the central plateau of the island (>1000 m altitude). There is little contemporary forest across this ecoregion [19], and it is currently dominated by grasses and shrubs. The eastern lowland

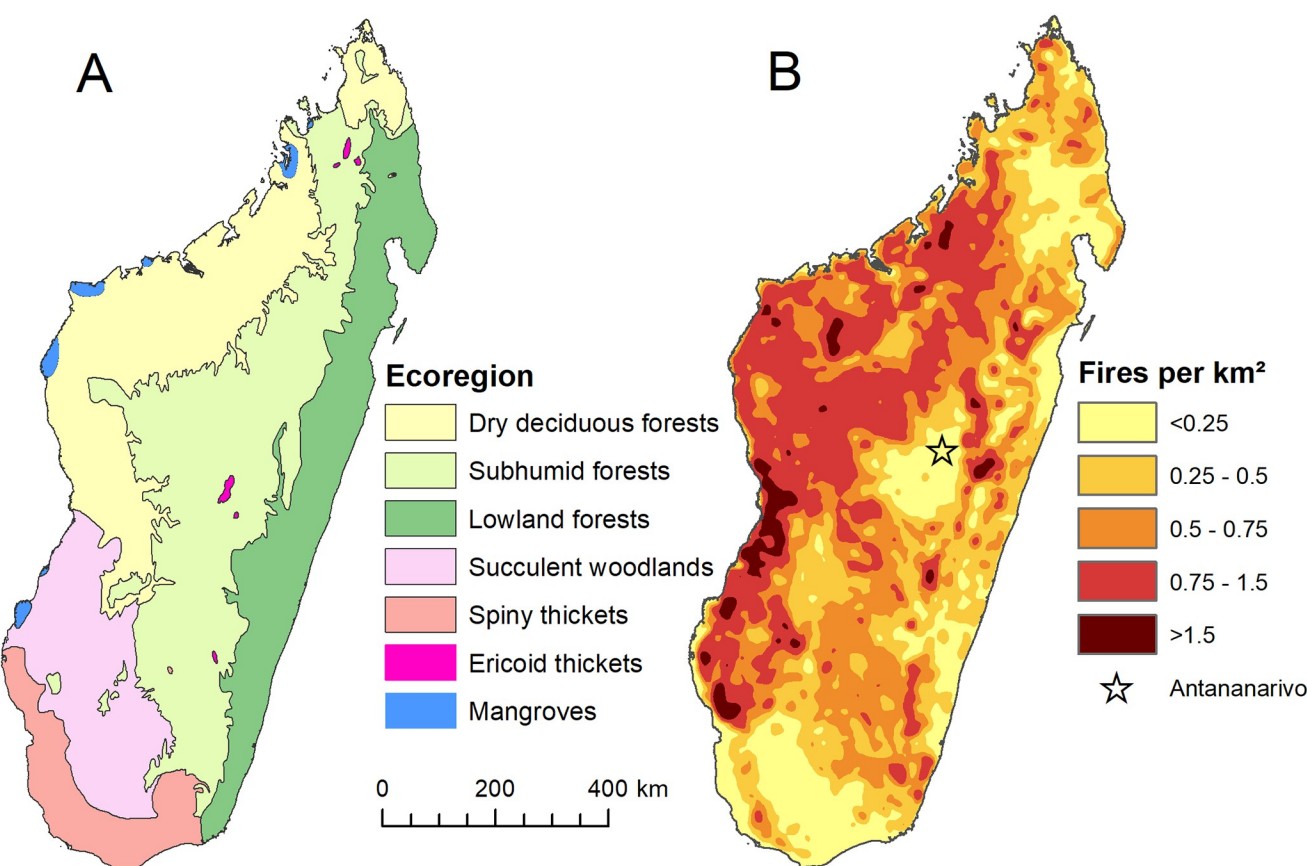

**Fig 1. Madagascar ecoregions and annual fire frequency.** (A) The ecoregions of Madagascar, based on WWF ecoregion classifications [37] Republished from [37] under a CC BY license, with permission from WWF. (B) Average annual fire frequency (number of fires per km$^2$ per year) from 2012–2019 VIIRS fire detection data with 1 km resolution.

forests ecoregion contains humid rainforest, with a much shorter dry season from approximately September to December. Across the island, dry season length tends to increase towards the west coast. The southwestern succulent woodlands and spiny thickets both have longer dry seasons lasting from approximately May to November. Towards the southwest, the climate is more arid, understory growth is less continuous, and vegetation is increasingly dominated by aphyllous shrubs, particularly in the southernmost spiny forests. Ericoid thickets are montane, shrub-dominated habitats found only in small areas above ~2000 m altitude [34]. Here, dry seasons are less pronounced, and habitats are relatively isolated from anthropogenic disturbance, meaning this may be the only ecoregion where fires are predominantly caused by lightning strikes rather than anthropogenic origins [34]. Mangroves are found in patches of coastal, estuarine forest along the west coast of Madagascar, surrounded by dry deciduous forests and succulent woodlands ecoregions.

Madagascar is home to stunning levels of microendemism across taxa [38], meaning that all ecoregions have unique species compositions and conservation outlooks. Mangroves, Tropical and Subtropical Dry Broadleaf Forests, and Tropical and Subtropical Moist Broadleaf Forests are all found in Madagascar, and have been identified as the three most fire-sensitive habitat types globally [29]. Here, we present the first quantification of fire occurrence across Madagascar using VIIRS fire detection. The objectives of our research were to 1) quantify fire frequency across Madagascar and how it is changing over time, 2) determine which ecoregions are most impacted by fire use and the fire seasonality within each ecoregion, and 3) assess the proximity to fire of remaining forests in each ecoregion and whether conservation efforts in national parks are reducing the fire pressure on protected forests.

## 2. Methods

### 2.1 VIIRS data

Fire data were obtained from the VIIRS (Visible Infrared Imaging Radiometer Suite) 375 m-resolution dataset (VNP14IMGT), which are managed by NASA and NOAA, and available to the public at https://firms.modaps.eosdis.nasa.gov/download/. The VIIRS instrument, launched in October 2011, orbits the Earth onboard the Suomi-National Polar-orbiting Partnership (S-NPP) satellite, detecting thermal anomalies/fire hotspots within 375 m pixels. There is currently a second VIIRS instrument orbiting Earth onboard the NOAA-20 satellite, but data collection did not begin until Jan 2020, which was outside of our study period. The VIIRS 375 m satellite fire-detection algorithm uses five single-gain channels, primarily using the brightness temperature of the infrared I4 (3.55–3.93 μm) and I5 (10.5–12.4 μm) channels for fire detection/identification, and channels I1 (0.6–0.68 μm), I2 (0.846–0.885 μm), and I3 (1.58–1.64 μm) for discrimination of clouds, water, and sun glint. Data were downloaded for Madagascar for 2012–2019. During this time frame, the VIIRS S-NPP satellite scanned the island for fires approximately every 12 hours, at ~10:30am and ~10:30pm, allowing for daytime and nighttime fire detection. For the purposes of this paper, the term "fire" will be used to refer to each detected 375 m fire hotspot. A single burning front can result in multiple fire hotspots, but to do so the front must extend multiple 375 m pixels at the same time of satellite overpass or burn for longer than ~12 hours (the return time of the satellite).

VIIRS data undergo onboard preprocessing and post-processing by NASA's LANCE (Land, Atmosphere Near-real-time Capability for EOS) to maximize the accuracy of fire detections and locations. The refined methods and the main I4 band's low saturation temperature of 367 K make this the most accurate and comprehensive fire detection database currently available. The VIIRS system is able to detect as many as 5-10x more fires than the MODIS system [39–41], and global estimates of false positive rates for nominal confidence pixels are only

0.03% of detected fires [42]. Global false positive estimates for low confidence fires are ~6%, and these false positives are clustered in industrial areas with highly reflective buildings, particularly in Eastern China [42]. Analysis of low confidence fire pixels in southern Africa showed they represented real fires of lower size/intensity, often bordering higher intensity fires [42]. False positives from industrial areas are not expected to have a notable impact on our study for two reasons: 1) there are few industrial areas in Madagascar, and 2) these areas tend to be close to cities and thus not near the remaining forests analyzed here. Low confidence pixels represent ~11% of all fires detected globally, and ~15% of fires within Madagascar. To obtain the most complete analysis possible of fire locations within Madagascar, all VIIRS fire detection in Madagascar were downloaded and analyzed. Locations of low confidence pixels were also examined visually to confirm that they corresponded to the locations of fire use indicated by nominal and high confidence pixels.

## 2.2 VIIRS processing/analysis

VIIRS data were downloaded in csv format and imported as points into ArcGIS ArcMap. A kernel density map at a resolution of 1 km was generated using all VIIRS points from 2012–2019. Change over time in fire frequency was calculated by inputting 5 km resolution kernel density maps for each year 2012–2019 into the Curve Fit tool [43] which fit a linear regression to the number of fires by year in each 5 x 5 km pixel. The output raster contained the average rate of change (slope) for the number of fires per year for every pixel on the map. To standardize these values by the background fire frequency, rate of change values were divided by the average fire density across the study period (2012–2019) and multiplied by 100 to convert the rates of change into a percentage. As such, the shown rates of change in percent represent a constant rate of change (as a % of the 2012–2019 average fires/km$^2$), and not an exponential/compounded rate of change. The ecoregion map was obtained from the World Wildlife Fund [37], with minor shapefile modifications made following Steffens et al. [44]. All fire points were spatially joined to the ecoregion they occurred within. Due to slight discrepancies in the detailed mapping of coastline between the ecoregions map and other datasets, 0.02% of total fires and 0.5% of total forest was not included within any ecoregion. All analyses were performed using the WGS 1984 UTM Zone 38S projection, except for area calculations, which used the Laborde projection.

## 2.3 Tree cover and conservation

Tree cover data were obtained at a scale of 30 m pixels from Vieilledent et al. [19]. The most recent dataset available shows estimated tree cover in 2017. These data were compared to the VIIRS detected fires in 2017 and the ecoregion boundaries to determine how threatened the remaining forest is by fire in each ecoregion. Buffers were created for 500 m, 750 m, 1 km, 1.5 km, 2 km, 2.5 km, and 3 km distances around all VIIRS fire points, and the proportion of forest in each ecoregion falling within these buffer distances was calculated in ArcMap. These values were also calculated separately for forest within the borders of a national park. National park boundaries were obtained from the Protected Planet database [45] in July 2019.

## 3. Results

### 3.1 Distribution of fires

VIIRS detected an average of 356,189 fires per year from 2012–2019, averaging 0.604 fires/km$^2$ per year. We found that 90.9% of fires were detected during the day, with only 9.1% detected at night. Daytime fire detection time ranged from 9:15 AM to 11:50 AM and averaged 10:32

AM. Nighttime fire detection time ranged from 9:00 PM to 11:15 PM. Average time of detection for night fires was 9:29 PM, as later satellite overpasses tended to detect fewer fires.

Average density of fires was highest in western Madagascar, particularly in the dry deciduous forest ecoregion, and in the northern part of the succulent woodlands ecoregion (Fig 1). During 2012–2019, average fire density in some locations reached as high as 4.1 fires/km² per year, with the highest values recorded northwest of the spiny thickets ecoregion, corresponding to the area immediately northeast of the city of Toliara. The largest area of fire frequency above 1.5 fires/km² was around the city of Morondava, encompassing both the Menabe Antimena Protected Area and Kirindy Mitea National Park. Regions of high fire frequency in the east were predominantly located around the last remaining strip of humid forest along the central escarpment (Fig 2B).

There were very few areas that were not affected by fire. In central Madagascar, the largest fire-free area was in the urban region surrounding the capital city of Antananarivo, extending down to another large city of Antsirabe. Other large areas of limited fire use were in the

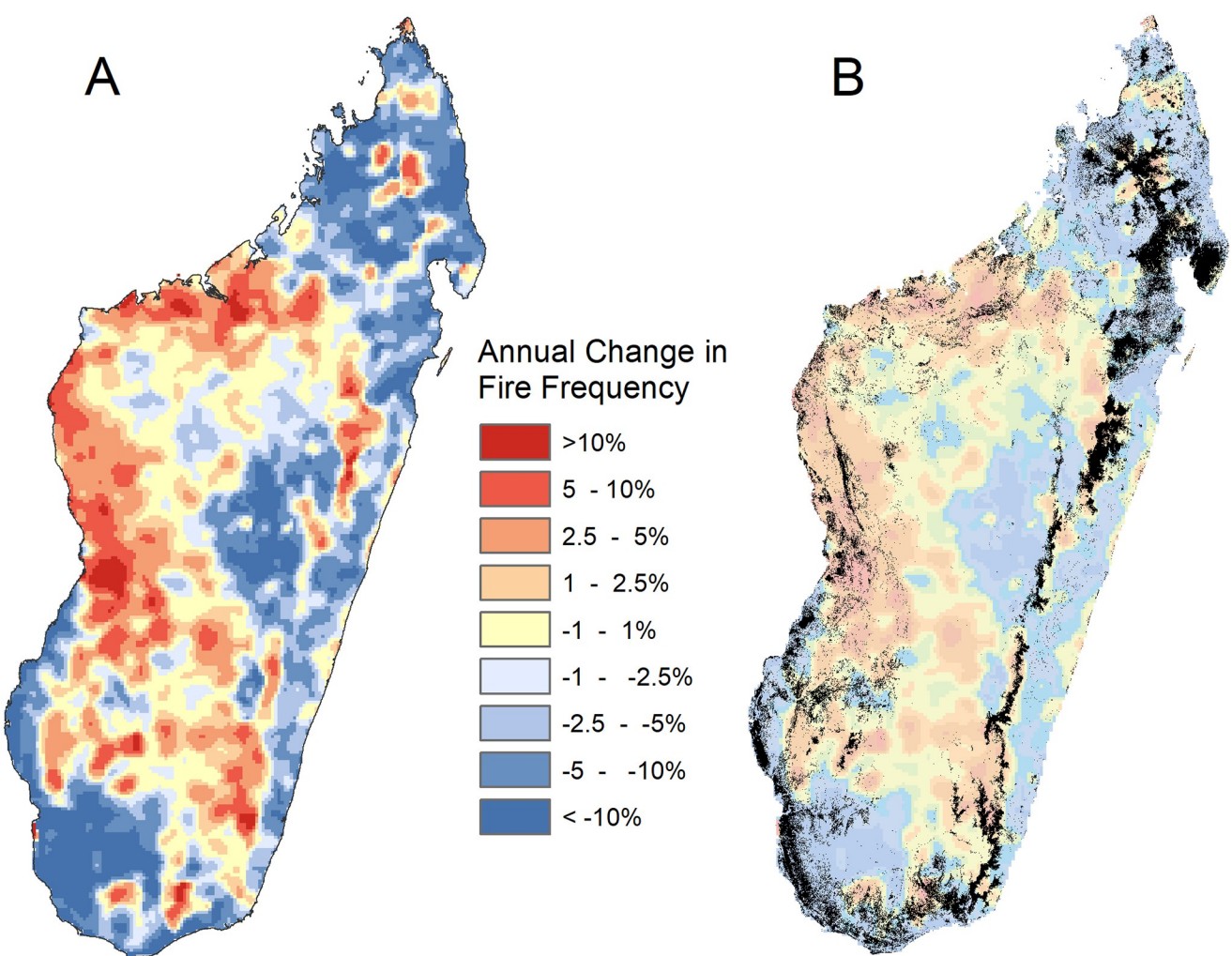

**Fig 2. Rates of change in fire frequency and remaining forest cover.** (A) Rate of change in VIIRS fire frequency for each 5 km x 5 km pixel from 2012–2019, expressed as a percentage of the average fire frequency (fires per km²) for that pixel in the 2012–2019 period. (B) Madagascar's remaining forest cover (in black) from 2017 from Vieilledent et al. [19] superimposed onto the data shown in (A) of change in fire frequency over time.

southwest of the island, in the spiny thickets and southern succulent woodlands, and in the northeast, near Masoala and Makira protected areas. Fire use was relatively low on the Masoala peninsula, but fire frequencies on the unprotected east coast of the peninsula were more than double than those inside the adjacent Masoala National Park. Areas of low fire use can also be seen along the eastern coast, a region with high population density [46], particularly around the city of Toamasina.

## 3.2 Changes in fire frequency over time

Across the entire island, there was no clear change in the total number of VIIRS detected fires over the study period (2012–2019), and MODIS fire detection data also suggested that the total number of fires island-wide has remained fairly constant since 2003 (S1 Fig). However, there were distinct regional changes in fire frequency (Fig 2). From 2012–2019, fire frequencies increased in the dry deciduous forests ecoregion and the southern portion of the central plateau/subhumid forests ecoregion. We found that 65% of land in the dry deciduous ecoregion saw a trend of increasing fire use from 2012–2019, compared to 49% of land in mangroves, 40% in subhumid forests, 38% in succulent woodlands, 23% in ericoid thickets, 16% in lowland forests, and 13% in spiny thickets ecoregions. Most of the lowland forests ecoregion saw decreases in fire frequency, but areas of distinct increases corresponded to the location of the few remaining lowland forests (Fig 2B). There were also increases in fire frequency around most of the remaining dry forest and some areas of northern humid forest.

## 3.3 Fire seasonality and frequency by ecoregion

During 2012–2019, the dry deciduous forests ecoregion recorded the highest average fire frequency with 0.80 fires/km$^2$ per year, followed by succulent woodlands (0.67), subhumid forests (0.59), mangroves (0.57), lowland forests (0.41), spiny thickets (0.40), and ericoid thickets (0.13) (Table 1). Most fires occurred in the late dry season, with 69.3% of all fires occurring between August and November. Fire seasons in the western ecoregions started earlier than those further east, with all ecoregions except for lowland forests and ericoid thickets seeing increases in fire use immediately after the end of the western wet season in April (Fig 3). Fire frequencies in dry deciduous forests, subhumid forests, and succulent woodlands increased steadily throughout the dry season (April—October) until the beginning of rain onset in late November, with very low fire frequency by December. Lowland forests and ericoid thickets did not experience increases in fire frequency until the end of the eastern wet season in August.

**Table 1. Summary of fire occurrence and proximity to remaining forest within each ecoregion.**

| Ecoregion | Total Area (km$^2$) | Remaining Forest (km$^2$) | Total Fires per Year | Annual Fires per km$^2$ | % Forest within 500m of fire | % Forest within 1km of fire | % Forest within 2km of fire |
|---|---|---|---|---|---|---|---|
| *Dry deciduous forests* | 150,824 | 14,696 | 120,158 | **0.80** | 32.0 | 60.5 | 84.5 |
| *Subhumid forests* | 198,745 | 15,195 | 116,690 | **0.59** | 13.2 | 30.8 | 57.5 |
| *Lowland forests* | 111,396 | 26,191 | 45,801 | **0.41** | 12.9 | 32.2 | 59.7 |
| *Succulent woodlands* | 79,402 | 9,989 | 53,199 | **0.67** | 32.0 | 57.9 | 81.8 |
| *Spiny thickets* | 43,097 | 16,369 | 17,101 | **0.40** | 12.4 | 28.7 | 53.8 |
| *Ericoid thickets* | 1,273 | 392 | 162 | **0.13** | 4.3 | 10.2 | 24.7 |
| *Mangroves* | 5,113 | 1,177 | 2,920 | **0.57** | 29.6 | 51.9 | 69.6 |
| *TOTAL* | **589, 850** | **84, 460** | **356, 189** | **0.60** | **18.6** | **39.3** | **64.9** |

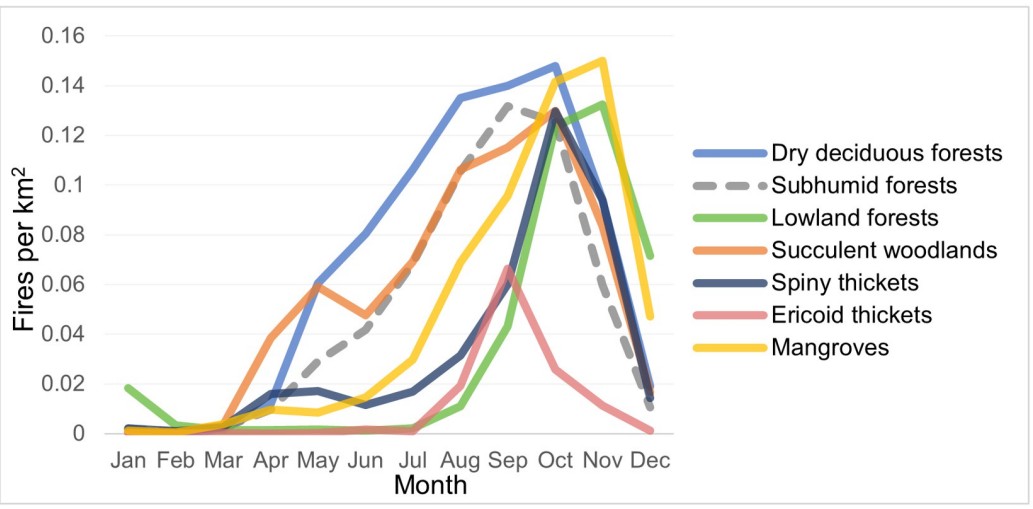

**Fig 3. Seasonality of fire frequency by ecoregion.** The average number of VIIRS fires detected per month in each ecoregion (expressed in the number of fires per km$^2$ per month) over the 2012–2019 time period.

However, fires in lowland forests rapidly increased from September-November to reach similar frequencies as the eastern ecoregions. The fire season in lowland forests lasted later into the year, with fire use not fully decreasing until February, relating to the later onset of rains in the eastern lowlands [47]. 80% of fires in the lowland forest ecoregion occurred from October to December. The spiny thickets ecoregion had slight increases in fire use starting in April, but fire use increased precipitously in October, with 56% of annual fires occurring in just October and November. Surprisingly, the highest average fires per km$^2$ for any month in any ecoregion was November in Mangroves, with 0.150 fires/km$^2$ in one month.

Values for total fires and fires/km$^2$ calculated using the annual averages from 2012–2019. Ecoregion boundaries based on the WWF ecoregion map [37]. Percent values for distance to fire calculated using the distance of each forest pixel to the nearest fire using 2017 VIIRS fire data and 2017 forest cover from Vieilledent et al. [19].

## 3.4 Fires around remaining forest

In 2017, of the remaining 84, 460 km$^2$ of forest on the island, 18.6% (15,681 km$^2$) was within 500 m of a fire, 39.3% (33,169 km$^2$) was within 1 km, and 64.9% (54, 774 km$^2$) was within 2 km. Forests in certain ecoregions were notably more threatened than others (Table 1). More than one quarter of remaining forest in the most threatened ecoregions, dry deciduous forests, succulent woodlands, and mangroves, was within 500 m of a fire in 2017 (32.0%, 32.0%, and 29.6%, respectively), and more than half was within 1 km (60.5%, 57.9%, and 51.9%).

Of the 8, 293 km$^2$ forest within national parks, only 7.1% (585 km$^2$) was within 500 m of a fire, 17.1% (1417 km$^2$) was within 1 km, and 38.6% (3200 km$^2$) was within 2 km. The proportion of national park forest within any distance to fire was lower than the island average for all ecoregions and all distance classes (Fig 4 and Table 2). However, the sample size of forest within a national park for mangroves and ericoid thickets were both small (< 15 km$^2$ of forest). Total fire frequency was 65% lower within national parks than the fire frequency within a 2 km buffer area surrounding each park, and 58% lower than their respective ecoregion average. Land within 2 km of the border of a national park had fire frequencies that were an average of 24% higher than their respective ecoregion averages.

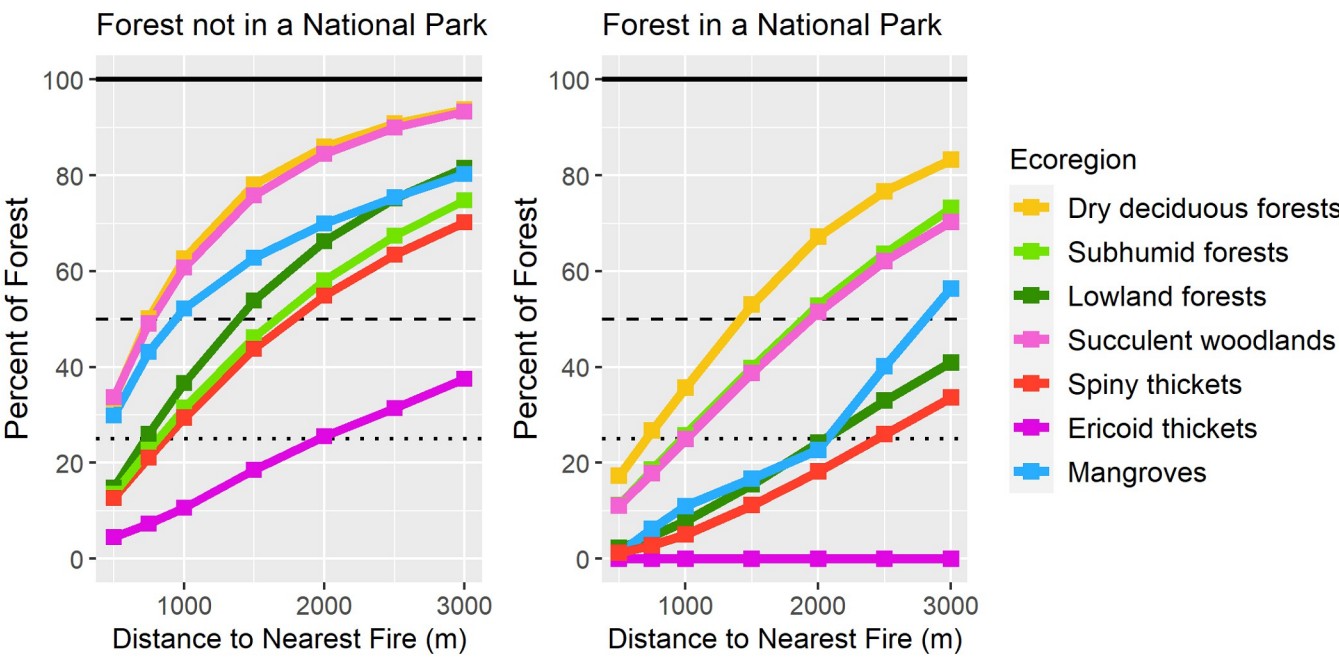

**Fig 4. Distance of remaining forest to the nearest fire.** The percent of all remaining forest across the island not within a national park (left) and exclusively within a national park (right) that was within each buffer distance of the closest fire in 2017. Values calculated using VIIRS 2017 fire detections and 2017 forest cover from Vieilledent et al. [19].

The proportion of remaining forest within each distance of the closest fire within 2017, split by whether or not the forest was protected by a national park. Values calculated using VIIRS 2017 fire detections and 2017 forest cover from Vieilledent et al. [19].

## 4. Discussion

### 4.1 Patterns of fire occurrence

Average fire frequency was consistently high throughout the dry deciduous forests ecoregion, reflecting the frequent use of fire as a tool for pasture maintenance in western Madagascar [22,

**Table 2. Differences in distance from forest to nearest fire by national park status.**

| Ecoregion | % of Total Forest within a National Park | Forest not within a National Park | | | Forest within a National Park | | |
|---|---|---|---|---|---|---|---|
| | | % Forest within 500m of fire | % Forest within 1km of fire | % Forest within 2km of fire | % Forest within 500m of fire | % Forest within 1km of fire | % Forest within 2km of fire |
| *Dry deciduous forests* | 8.30 | 33.3 | 62.7 | 86.1 | 17.3 | 35.6 | 67.2 |
| *Subhumid forests* | 11.16 | 13.5 | 31.5 | 58.0 | 11.1 | 25.8 | 52.8 |
| *Lowland forests* | 15.58 | 14.9 | 36.7 | 66.3 | 2.2 | 7.9 | 24.2 |
| *Succulent woodlands* | 7.91 | 33.8 | 60.8 | 84.4 | 11.1 | 25.0 | 51.5 |
| *Spiny thickets* | 2.89 | 12.7 | 29.4 | 54.9 | 1.2 | 5.1 | 18.2 |
| *Ericoid thickets* | 3.44 | 4.5 | 10.6 | 25.6 | 0.0 | 0.0 | 0.0 |
| *Mangroves* | 0.90 | 29.9 | 52.2 | 70.0 | 1.1 | 10.9 | 22.7 |
| *TOTAL* | **9.82** | **19.8** | **41.7** | **67.7** | **7.1** | **17.1** | **38.6** |

26]. The high fire frequencies surrounding remaining forests in the east are likely indicative of slash-and-burn agriculture that has already destroyed more than 50% of Madagascar's eastern rainforests [19]. The quantification of fire use presented here confirms previous identifications of the most fire-threatened ecosystems. The forests surrounding Toliara are believed to be under heavy fire pressure from charcoal production [31], and agricultural burning is a major threat to conservation efforts in Kirindy-Mitea National Park, where 6% of all remaining forest burned within a single dry season [48].

The northeastern humid forests around Masoala and Makira protected areas represent the largest remaining sections of intact forest on the island (Fig 2B), so it is reassuring for conservation efforts to see infrequent fire use across this region. However, the visible increases in fire frequency outside the borders of Masoala National Park suggest that these forests still may become threatened by fire use, and that national parks may help mitigate this damage. The lack of fire use in the large area surrounding Antananarivo, however, is likely the result of minimal vegetation remaining to burn rather than a natural ecosystem free from fire use. In the southwest, the large area of minimal burning may be due to a lack of continuous flammable fuels to enable fire spread in this arid vegetation type.

## 4.2 Changes in fire use over time

Analysis of the rate of change in fire frequency between 2012 and 2019 revealed that fire frequencies are increasing throughout the dry deciduous forests ecoregion, and disproportionately around the last remaining strip of humid lowland forests that runs north-south along the edge of the central escarpment/subhumid forests ecoregion. The forests in these areas are already subject to fire frequencies that are likely unsustainable for tropical forest. Research from mainland Africa suggests increasing population size decreases fire use in areas of high population density, but increases fire use in areas of low population density [49]. Population density in Madagascar is believed to be highest around Antananarivo, and along the eastern coast [46], which largely saw decreases in fire frequency. Population density is believed to be lowest across the western dry deciduous forests ecoregion and the southern central plateau/ subhumid forests ecoregion [46], which saw increases in fire frequency. While high population densities have reduced total burned area worldwide [50], this is usually the result of the conversion of natural, burnable land into agricultural or urban land, and does not necessarily indicate reduced fire pressures on remaining habitat. As populations in Madagascar continue to expand [33], the heavy reliance of local people on the land for agriculture [22] and extractive resources (primarily wood fuel) [51] will continue to increase the fire pressures on remaining forests, which can be seen around the remaining eastern humid forest.

## 4.3 Ecoregion fire regimes

Considerable fire use pervaded all of Madagascar's ecoregions. Total annual fire use was highest in the western dry deciduous forests ecoregion, and second highest in succulent woodlands, despite a large area of minimal fire use in the south of the ecoregion. Ericoid thickets were the only ecoregion with low average fire frequency (3 to 6 times lower than any other ecoregion) and a fire season that did not increase continuously throughout the dry season, supporting the idea that fires in ericoid thickets are predominantly started by lightning, as opposed to anthropogenic fires [34]. Fire seasonality in all other ecoregions peaked in the late dry season of each ecoregion. Fire season was most condensed in the lowland forests ecoregion, but here most fires represent deforestation that may have actually occurred months prior to biomass burning [22].

Fire frequency was surprisingly high in coastal mangroves, with the highest average single-month (November) fire frequency of any ecoregion. Mangrove forests mitigate erosion [52]

and provide resources and aquaculture opportunities to local people [53], but in Madagascar they are being harvested at unsustainable rates [54–56]. The noteworthy fire use in this semi-aquatic ecosystem reflects the high anthropogenic pressures that Madagascar's mangroves face. However, this analysis may not encapsulate all of Madagascar's mangroves, as fragments of mangrove forest also extend along the western coast of Madagascar, outside of the WWF's "Mangroves" ecoregion [19].

## 4.4 Implications of fire use for Madagascar's forests

Based on VIIRS fire detections, 40% of remaining forest in Madagascar was within 1 km of an active fire within a single year. The overuse of fire has already contributed to the extensive deforestation and habitat degradation that Madagascar has experienced over the past few decades [19, 20], and our analysis suggests that forests in Madagascar are still heavily threatened by anthropogenic fires. Slash-and-burn agriculture is widespread across the east [32] and west [30, 31] of Madagascar, and the cycles of fire use and crop growth deplete soil nutrients and drastically reduce the number of forest species [26, 57]. In grasslands, fires eliminate patches of scrub or forest cover that would otherwise reduce erosion [58] and spread into forest edges, leading to their conversion into secondary grassland [26]. Grassland fires can spread into forests as crown fires that result in the immediate mortality of canopy and understory vegetation, or more commonly as surface fires that only burn understory biomass [2]. However, the damage caused by surface fires can result in delayed mortality of more than one third of tree species within 3 years [59] and reduce canopy cover by 10–40% in humid forests [3, 11]. Reduced canopy cover increases fuel drying, and tree mortality increases fuel availability, which both create a positive feedback loop where subsequent fires are more intense and more likely to occur [7]. Once fires facilitate the conversion of forests into grasslands, the loss of soil nutrients and burning of newly established shoots make it near-impossible for the natural re-establishment of forests [19].

## 4.5 An underestimation of fire use

While these data show an alarming number of fires across all of Madagascar's ecoregions, VIIRS data likely underestimate the total number of fires across the island. The S-NPP satellite collected VIIRS data around 10:30am and 10:30pm each day, but most fires in Madagascar burn in the afternoon [60, 61], so the VIIRS data likely missed a considerable proportion of total fires. Afternoon fires would only be detected by these VIIRS data if they were started before ~10:30 am, or continued to burn into the night. Globally, only a small fraction of daily fires continues to burn overnight [61], and in these Madagascar data, nighttime detections represented only 9.1% of annual fires. This also means that few fire pixels would be counted twice by VIIRS, as it would require fire to burn into the night and for a minimum of 12 hours.

Surface fires may also be under-detected in these VIIRS data. Satellite fire detection datasets have struggled to detect small understory fires that are obscured by forest cover and/or smoke emissions [62, 63]; however, the low detection threshold of VIIRS should reduce these errors [64]. Reduced detection abilities may result in an underestimation of fires in forested regions compared to open grasslands, which could result in an underestimation of fires within national parks, which have disproportionately higher forest cover. Thus, if fires within forest are under-detected, the proportion of remaining forest impacted by fire is even greater than what is estimated here by VIIRS detections.

Cloud cover reduces the ability of VIIRS satellites to detect active fires [65] and would have contributed to the underestimation of the number of fires across Madagascar. This underestimation may have disproportionately affected regions or time periods with greater cloud cover,

particularly during the rainy season(s) or in the eastern lowland forests ecoregion where the rainy season lasts longer into the year. Thus, while the increased fuel moisture during the rainy season would reduce fuel flammability, estimates of fire use during periods of heavy cloud cover may be further deflated due to under-detection.

### 4.6 Conservation implications and future directions

Given the negative effects of fire occurrence on forest cover, it is reassuring for conservation outlooks to see here that fire frequencies within Madagascar's national parks were 58% lower than the averages for their ecoregions, and were an average of 65% lower than areas directly surrounding each park. The proportion of forest within 500 m of a fire was notably lower within national parks (7.1% inside vs 19.8% outside, 2.8 times lower), as was the proportion of forest within 1 km of a fire (17.1% vs 41.7%, 2.4 times lower). These results are in line with previous studies that suggest conservation efforts can reduce the number of fires across the tropics [66], and in individual regions of Madagascar [67]. However, most of Madagascar's protected areas are found outside of national parks in special reserves and community-managed protected areas [68]. These additional protected area types vary considerably in their ability to establish and enforce boundaries [68], and thus were not included in this analysis. Understanding the ability of all protected area types to protect their ecosystems from fires will require future analyses.

Conservation methods used to protect forests from fires vary considerably by region. In some protected areas surrounded by savanna (e.g., Ankarafantsika National Park), park managers use prescribed burns in the early dry season at the park's forest-grassland boundary to create fire breaks and reduce fuel loads for the late dry season (Steffens TS, personal communication) when fires pose the greatest risk to remaining forests [26]. Restricting fire use is often necessary for conservation, but complete bans on fire use that do not consider the larger factors leading to fire use, like poverty and the need to expand or rejuvenate agricultural lands, have been criticized for their limited effectiveness [22, 26]. However, conservation areas that prioritize the livelihoods of local communities have also been criticized for their limited ability to preserve ecosystems [35, 68–70]. Ultimately, research suggests that funding amount [67] and number of guards per km$^2$ of land [66] are the most important factors to protect remaining ecosystems from fires.

To fully understand the threat that rampant fire use poses to Madagascar's conservation efforts, we need more data on how different tree and faunal species respond to fire in their habitats. Studies of fire-related mortality of Madagascar's tree species are few [71], but research suggests that even tree species that evolved alongside fire are now threatened by too frequent burning [27]. Studies of faunal responses to fire in the tropics are incredibly limited [72, 73], particularly for primates which are one of Madagascar's top conservation priorities in lemurs. Fires decrease the production of fruits and flowers [12, 27, 74] which many lemurs rely upon, and can exacerbate hunting pressures by clustering species at surviving food trees [12]. Many lemur populations already face hunting at unsustainable levels [75, 76]. Fire can also create new forest edges [2, 77], contributing to Madagascar's issue of rampant forest fragmentation [19], which can isolate small, unsustainable populations [78] and negatively impact edge-averse species [79]. Fire has long been discussed as a contributing factor to the rapid decline of Madagascar's biodiversity [23–25], and we hope that this first large-scale quantification of the frequency and distributions of fires across Madagascar will help inform discussions on the current levels of fire use.

## Supporting information

**S1 Fig. Total number of fires detected per year in Madagascar by VIIRS and MODIS datasets.** Total number of fires detected by MODIS (left axis) from 2003–2019 and VIIRS (right

axis) from 2012–2019. VIIRS detected an average of 5.6 times more fires than MODIS.
(TIF)

**S1 Data.**
(ZIP)

**S2 Data.**
(ZIP)

**S3 Data.**
(ZIP)

**S4 Data.**
(ZIP)

**S5 Data.**
(ZIP)

**S6 Data.**
(ZIP)

**S7 Data.**
(ZIP)

**S8 Data.**
(ZIP)

**S1 File.**
(PDF)

## Acknowledgments

We are grateful for the NASA FIRMS system for making the VIIRS fire data publicly available, and for the WWF for allowing us to use their ecoregion data. We would also like to thank University of Toronto colleague Fernando Mercado Malabet for his assistance and expertise with spatial analysis in ArcGIS.

## Author Contributions

**Conceptualization:** Shawn M. Lehman.

**Formal analysis:** Tristan Frappier-Brinton.

**Funding acquisition:** Shawn M. Lehman.

**Investigation:** Tristan Frappier-Brinton, Shawn M. Lehman.

**Methodology:** Tristan Frappier-Brinton.

**Project administration:** Shawn M. Lehman.

**Resources:** Tristan Frappier-Brinton.

**Software:** Tristan Frappier-Brinton.

**Supervision:** Shawn M. Lehman.

**Visualization:** Tristan Frappier-Brinton.

**Writing – original draft:** Tristan Frappier-Brinton, Shawn M. Lehman.

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
