## [Decision Letter · Decision Letter 0]

21 Oct 2021

PONE-D-21-27734The burning island: Spatiotemporal patterns of fire occurrence in MadagascarPLOS ONE

Dear Dr. Lehman,

Thank you for submitting your manuscript to PLOS ONE. After careful consideration, we feel that it has merit but does not fully meet PLOS ONE’s publication criteria as it currently stands. Therefore, we invite you to submit a revised version of the manuscript that addresses the points raised during the review process.

We look forward to receiving your revised manuscript.

Kind regards,

Ji-Zhong Wan

Academic Editor

PLOS ONE

Journal Requirements:

2. Please provide the specific VIIRS data you extracted to use in the study, either by adding a URL to a data repository in the Methods section or by uploading a supplemental file

3.We note that Figure 1 in your submission contain [map/satellite] images which may be copyrighted. All PLOS content is published under the Creative Commons Attribution License (CC BY 4.0), which means that the manuscript, images, and Supporting Information files will be freely available online, and any third party is permitted to access, download, copy, distribute, and use these materials in any way, even commercially, with proper attribution. For these reasons, we cannot publish previously copyrighted maps or satellite images created using proprietary data, such as Google software (Google Maps, Street View, and Earth). For more information, see our copyright guidelines: http://journals.plos.org/plosone/s/licenses-and-copyright.

Additional Editor Comments (if provided):

Please revise the manuscript according to the reviewers' comments fully.

Reviewers' comments:

Reviewer's Responses to Questions

**Comments to the Author**

1. Is the manuscript technically sound, and do the data support the conclusions?

Reviewer #1: Yes

Reviewer #2: Yes

2. Has the statistical analysis been performed appropriately and rigorously? 

Reviewer #1: Yes

Reviewer #2: Yes

3. Have the authors made all data underlying the findings in their manuscript fully available?

Reviewer #1: Yes

Reviewer #2: Yes

4. Is the manuscript presented in an intelligible fashion and written in standard English?

Reviewer #1: Yes

Reviewer #2: Yes

5. Review Comments to the Author

Reviewer #1: This paper analyzes VIIRS satellite fire detection data to assess which regions of Madagascar have the most prevalent fire use, how fire use is changing over time, and what this means for Madagascar’s remaining forest ecosystems.

In my opinion, the topic is very interesting and worth investigating. I Have just a few minor comments:

- The abstract should be more assertive in describing the novelty of the paper;

- The abstract and Introduction should mention the methodology adopted in the research;

- There should be a small section for conclusions.

Reviewer #2: This study quantifies VIIRS hotspot detection, a proxy for fire use, on Madagascar, summarized by different ecoregions. It is well written and interesting There is no statistical modelling of drivers of fire use in different locations, which would provide further insight into fire use. However, the study provides an important summary/quantification of the issue, which sets the stage for greater future research. The authors are also clear about their method, their aim with the study and the discussion flows well from the results. The authors have summarized and presented the information well, and I believe this will be a valuable paper to researchers interested in human fire use, slash and burn agriculture and deforestation in the tropics.

I have several minor comments – mainly some clarification around certain text and some further explanation that is required to improve the manuscript - see attached document.

6. PLOS authors have the option to publish the peer review history of their article (what does this mean?). If published, this will include your full peer review and any attached files.

Reviewer #1: No

Reviewer #2: No

---

## [Author Response · Author response to Decision Letter 0]

7 Dec 2021

We would like to thank our reviewer for their knowledgeable and insightful feedback that greatly improved the clarity of our manuscript. Please find an itemized list below of how we integrated specific comments into our revised manuscript.

Figure 1 copyright concerns: There were concerns expressed in the feedback of our manuscript that the ecoregion map used in Figure 1 may be subject to previous copyright that would impede the publication of these data in PLOS ONE under the PLOS ONE’s CC BY license. However, the data used here were obtained from the World Wildlife Fund’s ecoregions map, which is available to the public to be used and edited under an active CC BY license. Slight edits to the borders of some ecoregion shape files were made following the regions used in the following article which SM Lehman also authored: 

Steffens TS, Ramsay MS, Mercado Malabet FM, Lehman SM. The Effects of Forest Loss and Fragmentation on non-Volant Mammals in Madagascar. In: Goodman SM, editor. The New Natural History of Madagascar. Princeton: Princeton University Press; Forthcoming.

The editing of these creative commons data obtained from the WWF does not change the copyright, and this Steffens et al. article was simply cited so that any edits of the WWF CC BY data can be tracked accordingly. We apologize that this was not made clear in the text of our manuscript, and we have made revisions to help clarify this in our Figure 1 caption and the section of the Methods describing the ecoregions map.

 The WWF Ecoregion map/data can be obtained online at: https://databasin.org/datasets/68635d7c77f1475f9b6c1d1dbe0a4c4c/

General comments 

These are my main comments to the authors. 

• In my opinion, you need to be clear how you are defining a “fire” in this paper. It looks like you are saying one hotspot is one fire, but one fire is usually defined (in fire behaviour/fire history studies) as the single continuous burnt area. That is, one wildfire can have multiple hotspots detected at once, e.g. in the picture below (VIIRS image overlayed with hotspots) I would say the green/yellow polygon (the eventual fire boundary) is one fire, and (at this particular time) there are multiple hotspots. So, I’d say you need to clearly define how you are using the word “fire”, or change your use of fire to hotspot or another term in the text and in table 1 for example. 

Response: We would like to thank the reviewer for pointing out that we did not adequately clarify this key definition to our paper. We have added a clearer description of how we will be using the term “fire”, and the details of VIIRS fire hotspots to Methods section 2.1 (Line 140)

• Adding some explanation about how clouds may have affected the results is needed. Are there any areas/ecoregions of Madagascar that a consistently cloudier than other areas? If so, you could expect that VIIRS detection would be lower here because clouds may shielded fires from detection by VIIRS. This may be important for the results, e.g. if the lowlands are cloudier, than VIIRS may under-detect fires in this region. I think some acknowledgement of this issue in the methods or in section 4.5 is warranted. 

Response: This is an excellent point, and we have added a brief paragraph to Section 4.5 (line 379) describing how cloud cover may disproportionately affect VIIRS detections at certain locations or in certain times of year.

Specific comments

Line 42: It would be useful to add some specific examples of how or why fires have negative impacts on tropical forests (e.g. trees easily killed by fire because they are not adapted), rather than only the broad statement about “devastating effects” etc. 

Response: Example added in text.

Line 43: “the most fire-threatened continent on Earth”. I’d add this to the next paragraph, as the previous paragraph doesn’t provide support for this statement. 

Response: Changed

Line 100 and 102: What does “ca.” represent? 

Response: Changed “ca.” to “approximately”

Line 102: The sentence starting “In the southwest, the climate becomes” has unclear wording. It sounds like the aphyllous shrubs continue south. I suggest rewording this sentence. 

Response: Changed

Line 125: (Section 2.1) Need to add a sentence or few words saying that there are two VIIRS instruments on separate satellites: Suomi NPP and NOAA-20. It is stated that you only used S-NPP, but it is worth saying there is another satellite now, but it is newer and had not collected data for your study period. 

Response: Clarification added in text.

Line 134: Similar to the previous point, better to clarify which satellite instrument. “VIIRS satellite” could refer to either, better to say VIIRS S-NPP or something similar.

Response: Changed

 Line 144: Were there any industrial/urban areas false detection that needed to be excluded for your study area? Perhaps this was accounted for anyway when doing the intersection with the ecoregion polygons 

Response: Addressed in line 153

Line 236: Just a question, could this later peak be because mangroves take a long time to dry out, but when they do they are very flammable? 

Response: This is an interesting question, and one that we hope to future studies to determine the reasons behind the observed patterns in fire occurrence across ecoregions.

Line 239: I believe the fig 3 caption should say “expressed in the number of fires per km2 per month) 

Response: Changed

Line 265: I found this chart a little confusing. This may be better as a bar chart i.e each buffer distance along the x axis and % along the y axis, and then a small bar for each ecoregion (repeated at each buffer distance). E.g. the “grouped bar chart” here: https://www.r-graph-gallery.com/48-grouped-barplotwith-ggplot2.html. Otherwise, add the points that make up each line so the buffer categories are clear.

Response: We have added points to the figure that denote each buffer distance, as well as horizontal lines at 100%, 50%, and 25% of remaining forests to make the differences between plots more clear and easier to interpret. We believe that the number of buffer distances included in the analysis would make a bar chart too expansive to include for all seven ecoregions, and a line graph allows for more data to be conveyed. We believe this is much clearer with the suggested addition of points along each line.

 Line 267: Saying “that was within each buffer distance of the closest fire” would make this clearer. 

Response: Changed

Line 271: Saying “effect” in the Table 1 caption implies a statistical model, where this just seems to be showing differences in summary statistics. I think you should say “Differences in distance to nearest fire by National park status” or something similar. 

Response: Changed

Line 288: Reassuring in terms of what? 

Response: Clarification added “reassuring for conservation efforts”

Line 298: “Analysis of the rate of change in fire frequency between 2012 and 2019 revealed that fire frequencies are increasing throughout the dry deciduous forests ecoregion“ I suggest adding a table (or adding to an existing table) the mean change for each ecoregion type. I think this statement is just based of looking at the maps. It would be good to see some numbers. 

Response: We agree that seeing the numbers for this statement help to make it more compelling. We have added the percentage of land within each ecoregion that saw an increasing trend in fire use across the study period to the Results section 3.2 (line 223)

Line 360-366: As in the general comment, how does cloud influence this? Are the lowlands cloudier, so hotspots undetected there. 

Response: Clarified in an added paragraph in Discussion Section 4.5

Also, you should not use the word “alarming” without explaining why. I would reword this final sentence (Line 366) to something like: If fires within forest are underestimated, the impact of fire on the forests is greater than can be determined/estimated by VIIRS detections..”, or something more similar. Response: Changed

Line 369: Avoid “reassuring” without further explanation. Or just explain that it is a positive in terms of conservation outcomes 

Response: Clarification added

Line 372: Is the 19.8% for non-national parks? Also probably better to say 2.8 times lower, rather than 2.8 fold decrease. 

Response: Clarified, and changed

Also, could it be national parks are more commonly in less fire-prone areas, e.g. steep/hilly areas more inaccessible to farmers, therefore less fires ignited there due to the terrain rather than national park status. Add a note if you think that could be influencing the results. 

Response: This is a very interesting topic, and one that we hope to explore more fully in future studies. Determining the reasons behind lower fire frequency within national parks will ideally require a pixel matching approach, but these methods are currently hindered in Madagascar by the lack of an accurate, high-resolution population density map that would allow habitats with comparable anthropogenic pressures, climate, and topography to be compared across protected statuses. Without such an analysis we do not know if topography/inaccessibility may be contributing to these results, and while some national parks along the eastern escarpment do have steep topography, national parks across the north and west are largely located on flatter land below the Central Plateau and close to the coast.

Line 389: What is guard density? Could you please clarify.

Response: Clarification added

---

## [Decision Letter · Decision Letter 1]

18 Jan 2022

The burning island: Spatiotemporal patterns of fire occurrence in Madagascar

PONE-D-21-27734R1

Dear Dr. Lehman,

We’re pleased to inform you that your manuscript has been judged scientifically suitable for publication and will be formally accepted for publication once it meets all outstanding technical requirements.

Kind regards,

Ji-Zhong Wan

Academic Editor

PLOS ONE

Additional Editor Comments (optional):

Reviewers' comments:

Reviewer's Responses to Questions

**Comments to the Author**

1. If the authors have adequately addressed your comments raised in a previous round of review and you feel that this manuscript is now acceptable for publication, you may indicate that here to bypass the “Comments to the Author” section, enter your conflict of interest statement in the “Confidential to Editor” section, and submit your "Accept" recommendation.

Reviewer #1: All comments have been addressed

Reviewer #2: All comments have been addressed

2. Is the manuscript technically sound, and do the data support the conclusions?

Reviewer #1: Yes

Reviewer #2: (No Response)

3. Has the statistical analysis been performed appropriately and rigorously? 

Reviewer #1: Yes

Reviewer #2: (No Response)

4. Have the authors made all data underlying the findings in their manuscript fully available?

Reviewer #1: Yes

Reviewer #2: (No Response)

5. Is the manuscript presented in an intelligible fashion and written in standard English?

Reviewer #1: Yes

Reviewer #2: (No Response)

6. Review Comments to the Author

Reviewer #1: The authors improved their paper inline with previous comments from the reviewrs. I have no further comments.

Reviewer #2: (No Response)

7. PLOS authors have the option to publish the peer review history of their article (what does this mean?). If published, this will include your full peer review and any attached files.

Reviewer #1: No

Reviewer #2: No

---

## [Editor Report · Acceptance letter]

23 Mar 2022

PONE-D-21-27734R1 

The burning island: Spatiotemporal patterns of fire occurrence in Madagascar 

Dear Dr. Lehman:

I'm pleased to inform you that your manuscript has been deemed suitable for publication in PLOS ONE. Congratulations! Your manuscript is now with our production department. 

Kind regards, 

on behalf of

Dr. Ji-Zhong Wan 

Academic Editor

PLOS ONE